# Towards novel herbicide modes of action by inhibiting lysine biosynthesis in plants

Tatiana P Soares da Costa[1†]*, Cody J Hall[1†], Santosh Panjikar[2,3], Jessica A Wyllie[1], Rebecca M Christoff[4], Saadi Bayat[4], Mark D Hulett[1], Belinda M Abbott[4], Anthony R Gendall[5,6], Matthew A Perugini[1]*

[1]Department of Biochemistry and Genetics, La Trobe Institute for Molecular Science, La Trobe University, Bundoora, Australia; [2]Australian Synchrotron, ANSTO, Clayton, Australia; [3]Department of Molecular Biology and Biochemistry, Monash University, Melbourne, Australia; [4]Department of Chemistry and Physics, La Trobe Institute for Molecular Science, La Trobe University, Bundoora, Australia; [5]Department of Animal, Plant and Soil Sciences, AgriBio, La Trobe University, Bundoora, Australia; [6]Australian Research Council Research Hub for Medicinal Agriculture, Bundoora, Australia

**Abstract** Weeds are becoming increasingly resistant to our current herbicides, posing a significant threat to agricultural production. Therefore, new herbicides with novel modes of action are urgently needed. In this study, we exploited a novel herbicide target, dihydrodipicolinate synthase (DHDPS), which catalyses the first and rate-limiting step in lysine biosynthesis. The first class of plant DHDPS inhibitors with micromolar potency against *Arabidopsis thaliana* DHDPS was identified using a high-throughput chemical screen. We determined that this class of inhibitors binds to a novel and unexplored pocket within DHDPS, which is highly conserved across plant species. The inhibitors also attenuated the germination and growth of *A. thaliana* seedlings and confirmed their pre-emergence herbicidal activity in soil-grown plants. These results provide proof-of-concept that lysine biosynthesis represents a promising target for the development of herbicides with a novel mode of action to tackle the global rise of herbicide-resistant weeds.

*For correspondence:
t.soaresdacosta@latrobe.edu.au (TPSC);
Matt.Perugini@gmail.com (MAP)

†These authors contributed equally to this work

## Introduction

Our ability to provide food security for a growing world population is increasingly challenged by the emergence and spread of herbicide-resistant weeds. Resistance has now been observed to the most widely used classes of herbicides. This includes amino acid biosynthesis inhibitors such as chlorsulfuron, glufosinate, and glyphosate, which target enzymes in the biosynthetic pathways leading to the production of branched-chain amino acids, glutamine, and aromatic amino acids, respectively (*Gaines et al., 2020*; *Hall et al., 2020*). The impact of herbicide resistance is exacerbated by the lack of new herbicides entering the market in the past 30 years, especially those with new mechanisms of action (*Duke, 2012*). Nevertheless, the successful commercialisation of such herbicides provides proof-of-concept that targeting the biosynthesis of amino acids offers an excellent strategy for herbicide development (*Hall et al., 2020*). Amino acids are not only essential building blocks for protein biosynthesis, but they also play important roles in physiological processes that are critical for plant growth and development, such as carbon and nitrogen metabolism, in addition to serving as precursors to a wide range of secondary metabolites (*Hildebrandt et al., 2015*).

One amino acid biosynthesis pathway that remains largely unexplored for herbicide development is the diaminopimelate (DAP) pathway (*Figure 1*), which is responsible for the production of L-lysine (here after referred to as lysine) in plants and bacteria (*Figure 1*; *Hall and Soares da Costa, 2018*).

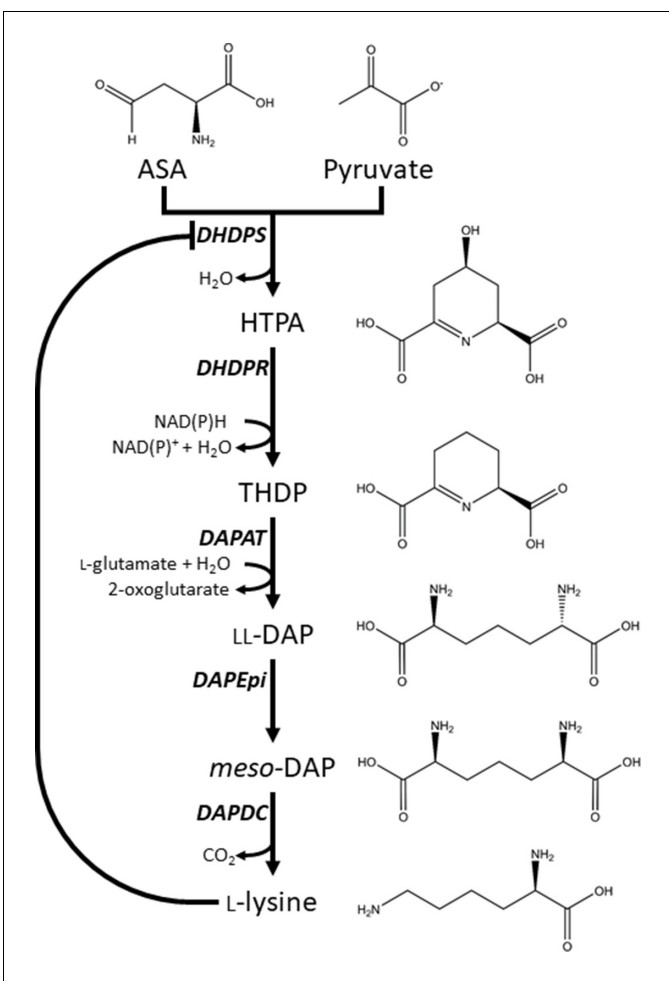

**Figure 1.** Lysine biosynthesis in plants. Plants utilise the diaminopimelate (DAP) pathway, a branch of the aspartate-derived super-pathway, to synthesise L-lysine. Firstly, L-aspartate semialdehyde (ASA) and pyruvate are converted to (4S)−4-hydroxy-2,3,4,5-tetrahydro-(2S)-dipicolinic acid (HTPA) in a condensation reaction catalysed by dihydrodipicolinate synthase (DHDPS). Dihydrodipicolinate reductase (DHDPR) then catalyses an NAD(P)H-dependent reduction of HTPA to produce 2,3,4,5-tetrahydrodipicolinate (THDP). THDP subsequently undergoes an amino-transfer reaction with L-glutamate, catalysed by diaminopimelate aminotransferase (DAPAT), to yield L,L-DAP. L,L-DAP is converted to *meso*-DAP by diaminopimelate epimerase (DAPEpi) and lastly, *meso*-DAP is decarboxylated by diaminopimelate decarboxylase (DAPDC) to yield L-lysine, which imparts a negative feedback loop on DHDPS.

Animals, including humans, do not synthesise lysine, and therefore, must acquire it from dietary sources (*Galili and Amir, 2013*; *Tomé and Bos, 2007*). Consequently, specific chemical inhibition of the DAP pathway in plants is unlikely to be harmful to animals and humans (*Hutton et al., 2007*). The DAP pathway commences with a condensation reaction between L-aspartate semialdehyde (ASA) and pyruvate to form (4S)−4-hydroxy-2,3,4,5-tetrahydro-(2S)-dipicolinic acid (HTPA), catalysed by HTPA synthase (EC 4.2.1.52), also known as dihydrodipicolinate synthase (DHDPS) (*Griffin et al., 2012*; *Soares da Costa et al., 2018*; *Soares da Costa et al., 2015*). HTPA is then reduced by dihydrodipicolinate reductase (DHDPR, EC 1.3.1.26) in the presence of NAD(P)H to produce 2,3,4,5-tetrahydrodipicolinate (THDP) (*Christensen et al., 2016*). In plants, THDP undergoes an amino-transfer by diaminopimelate aminotransferase (DAPAT, EC 2.6.1.83) to form L,L-DAP, which is converted to *meso*-DAP by diaminopimelate epimerase (DAPEpi, EC 5.1.1.7) (*Hudson et al., 2005*; *McCoy et al., 2006*). Lastly, *meso*-DAP is irreversibly decarboxylated by diaminopimelate decarboxylase (DAPDC, EC 4.1.1.20) to produce lysine (*Peverelli and Perugini, 2015*). Lysine regulates flux through the

pathway by binding allosterically to DHDPS and inhibiting the enzyme. Thus, DHDPS catalyses the rate-limiting step of the DAP pathway (*Geng et al., 2013*; *Soares da Costa et al., 2016*).

Due to the central role of DHDPS in lysine production in plants, this enzyme has been proposed as a potential target for the development of herbicides (*Griffin et al., 2012*; *Soares da Costa et al., 2018*). Indeed, the lysine analogue, *S*-(2-aminoethyl)-L-cysteine, halts rooting of potato tuber discs at mid-micromolar concentrations (*Perl et al., 1993*; *Ghislain et al., 1995*). However, given its poor *in vitro* potency against plant DHDPS, it is believed that this analogue inhibits plant growth by competing with lysine for incorporation into proteins rather than inhibition of the DHDPS enzyme (*Ghislain et al., 1995*; *Perl et al., 1993*). Plants typically have two annotated DHDPS-encoding genes (*DHDPS*) (*Figure 2—figure supplement 1*; *Craciun et al., 2000*; *Sarrobert et al., 2000*; *Vauterin and Jacobs, 1994*). In *Arabidopsis thaliana*, these genes are *At3G60880* (*DHDPS1*) and *At2G45440* (*DHDPS2*), which encode chloroplast-targeted AtDHDPS1 and AtDHDPS2, respectively (*Jones-Held et al., 2012*). RNA sequencing data have elucidated that both DHDPS-encoding genes are expressed at all stages of *A. thaliana* development, with the most prominent expression at the seed development stages, in the dry seeds and during germination (*Klepikova et al., 2016*). Interestingly, maximal *DHDPS2* expression is approximately 3-fold greater than that of *DHDPS1* (*Klepikova et al., 2016*). Moreover, upon comparison to the known glyphosate herbicide target, 5-*enol*pyruvyl-shikimate 3-phosphate synthase (*At1G48860*), expression of *DHDPS1* is considerably lower at almost all developmental stages, while expression of *DHDPS2* is comparable at all stages, except in the dry seeds (*Klepikova et al., 2016*). This is a key consideration in target validation as low expressing targets will require less inhibitor to achieve phytotoxicity. Double knockouts of *DHDPS1* and *DHDPS2* result in non-viable embryos even after exogenous supplementation with lysine, indicating that DHDPS activity is essential (*Jones-Held et al., 2012*). AtDHDPS enzymes exist as homotetramers (*Figure 2A*), with the active site located within the $(\beta/\alpha)_8$-barrel (*Figure 2B*) and the allosteric cleft in the tight-dimer interface located in the interior of the structure (*Figure 2C*; *Griffin et al., 2012*; *Hall et al., 2021*).

In this study, we describe a new class of plant DHDPS inhibitors. We show that these compounds display micromolar potency *in vitro* and *in planta* against *A. thaliana* using a combination of enzyme kinetics, seedling, and soil assays, whilst exhibiting no cytotoxic effects in bacterial or human cell lines at equivalent concentrations. Furthermore, we employ X-ray crystallography to show that these compounds target a previously unexplored binding site within DHDPS, which is highly conserved amongst plant species. Thus, this study provides proof-of-concept that lysine biosynthesis represents a promising pathway to target for the development of herbicides with a new mode of action and highlights a novel DHDPS binding pocket to assist in the discovery of herbicide candidates.

## Results

### High-throughput chemical screen for inhibitor discovery

A high-throughput screen of a library of 87,648 compounds was conducted against recombinant DHDPS enzyme by the Walter and Eliza Hall Institute High Throughput Chemical Screening Facility (Melbourne, Australia). The *o*-aminobenzaldehyde (*o*-ABA) colourimetric assay was used to estimate DHDPS activity via the formation of a purple chromophore that can be measured at 520–540 nm (*Yugari and Gilvarg, 1965*). Using a cut-off equal to the mean $\pm 3\times$ standard deviation for classification as a hit compound, 435 compounds out of 87,648 were identified as hits at 20 mM (hit rate = 0.50%). The activity of these 435 compounds was confirmed at the same concentration as the primary screen, resulting in 38 compounds demonstrating >40% inhibition of the DHDPS enzymatic reaction (confirmation rate = 8.7%). A counter screen was employed to exclude false-positive compounds i.e., compounds that interacted with *o*-ABA detection or absorbance quantification. The compounds that displayed confirmed DHDPS inhibition were subsequently progressed to full dose response titration assays using recombinant DHDPS. One promising compound from the screen was (*Z*)−2-(5-(4-fluorobenzylidene)−2,4-dioxothiazolidin-3-yl)acetic acid (FBDTA). The characterisation of two thiazolidinedione analogues containing methoxy substituents, MBDTA-1 and MBDTA-2, will be discussed here (*Figure 3*, *Supplementary file 1*).

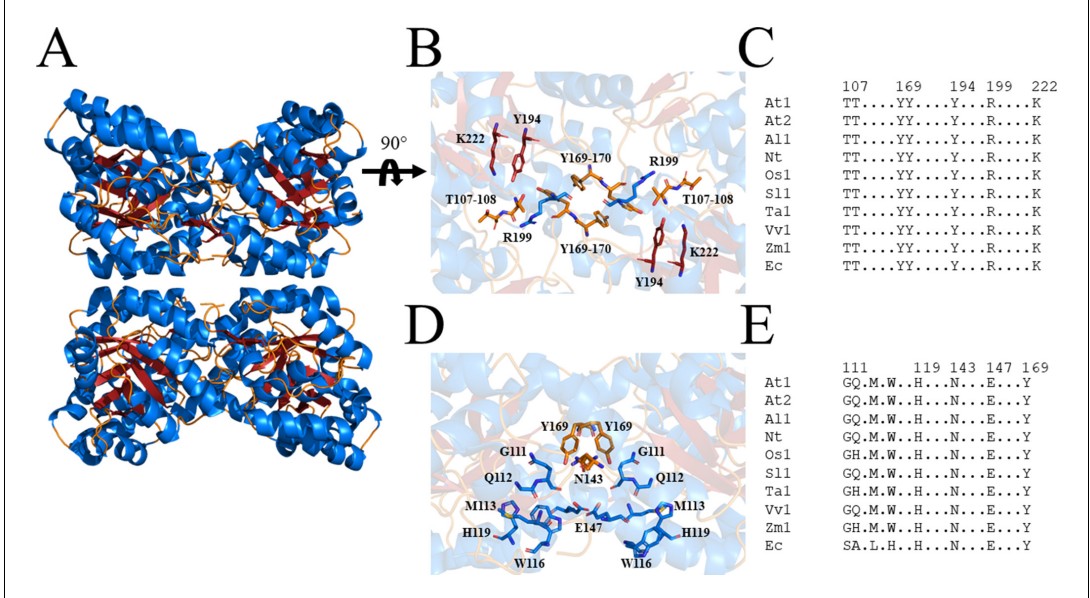

**Figure 2.** Structure and sequence conservation of plant DHDPS enzymes. (**A**) Cartoon structure of *Arabidopsis thaliana* (At) DHDPS1 (PDB: 6VVI) in the unliganded form illustrating the 'back-to-back' homotetramer conformation. (**B**) Cartoon structure of AtDHDPS1, with residues critical for catalysis shown as sticks. (**C**) Multiple sequence alignment of residues important for catalysis. (**D**) Cartoon structure of AtDHDPS1, with residues important for lysine binding and allosteric regulation shown as sticks. Residues are coloured by nitrogen (blue), oxygen (red), and sulfur (yellow). Images were generated using *PyMOL* v 2.2 (Schrödinger). (**E**) Multiple sequence alignment of residues involved in allosteric lysine binding. Sequences are AtDHDPS1 (At1; UNIPROT ID: Q9LZX6), AtDHDPS2 (At2; UNIPROT ID: Q9FVC8), *Arabidopsis lyrata* DHDPS1 (Al1; UNIPROT ID: D7LRV3), *Nicotiana tabacum* DHDPS (Nt; UNIPROT ID: Q42948), *Oryza sativa* DHDPS1 (Os1; UNIPROT ID: A0A0K0K9A6), *Solanum lycopersicum* DHDPS1 (Sl1; UNIPROT ID: A0A3Q7IMG0), *Triticum aestivum* DHDPS1 (Ta1; UNIPROT ID: P24846), *Vitis vinifera* DHDPS1 (Vv1; UNIPROT ID: A0A438E022), *Zea mays* DHDPS1 (Zm1; UNIPROT ID: P26259), and *Escherichia coli* (Ec) DHDPS (UNIPROT ID: P0A6L2). Residues are numbered according to AtDHDPS1 with dots (.) representing interspacing residues. Sequences were aligned in *BioEdit* (v 7.2.5) (**Hall, 1999**) using the *ClustalW* algorithm (**Thompson et al., 1994**). The online version of this article includes the following figure supplement(s) for figure 2:

**Figure supplement 1.** Sequence alignment of plant DHDPS enzymes.

## Efficacy of inhibitors on recombinant DHDPS

The inhibitory activity of MBDTA-1 and MBDTA-2 against both recombinant *A. thaliana* DHDPS proteins was quantitated using a DHDPS–DHDPR coupled assay (**Atkinson et al., 2013**). This was achieved by titrating different concentrations of each compound with substrates fixed at previously determined Michaelis–Menten constant values (**Griffin et al., 2012**; **Hall et al., 2021**). The $IC_{50}$ values of MBDTA-1 and MBDTA-2 against AtDHDPS1 were $126 \pm 6.50$ μM and $63.3 \pm 1.80$ μM, respectively (**Figure 4A**). Similarly, the dose–response curves for AtDHDPS2 yielded $IC_{50}$ values of $116 \pm 5.20$ μM and $64.0 \pm 1.00$ μM for MBDTA-1 and MBDTA-2, respectively (**Figure 4B**). As these compounds represent a novel class of inhibitors of plant DHDPS, we set out to assess the mechanism of inhibition by examining the binding of MBDTA-2 to DHDPS using X-ray crystallography.

## Molecular basis for inhibitor binding

To probe the molecular determinants for inhibition, recombinant AtDHDPS1 was co-crystallised with MBDTA-2 using the same crystallisation conditions as for the apo enzyme with the addition of inhibitor (**Hall et al., 2021**). Diffraction data were obtained at a maximal resolution of 2.29 Å, phases solved by molecular replacement and repeating rounds of model building and refinement were performed that allowed us to generate an atomic inhibitor-bound model (**Figure 5A**, **Table 1**). We initially found several MBDTA-2 molecules at the crystal contact formed between protein molecules at chains B and D. Specifically, two parallel MBDTA-2 molecules were bound to H187 (of the symmetry mate) and F210 with complete occupancy (**Figure 5—figure supplement 1**). However, given that these molecules were found solely at the crystal interface and were absent in chains A and C, they were assumed to be a result of non-specific binding.

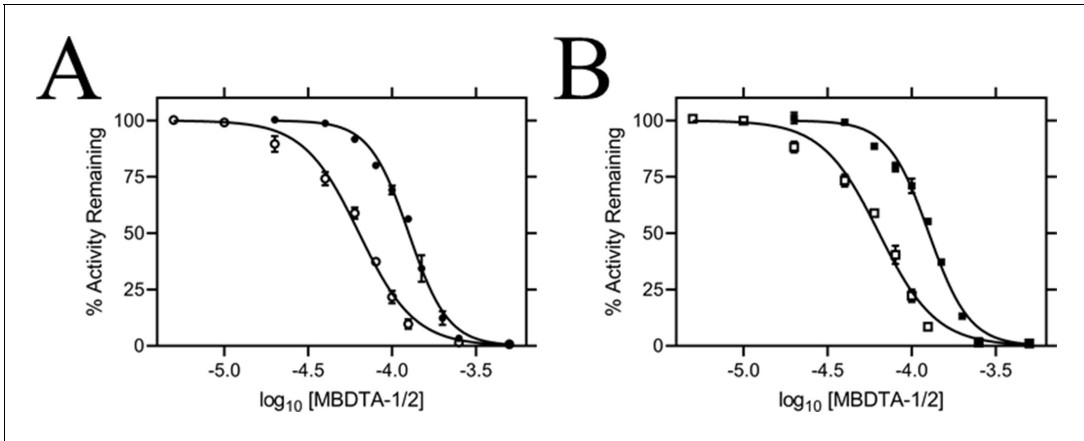

**Figure 3.** Structure of DHDPS inhibitors. Chemical structures of the hit compound, (Z)—2-(5-(4-fluorobenzylidene)—2,4-dioxothiazolidin-3-yl)acetic acid (FBDTA), and two analogues, (Z)—2-(5-(2-methoxybenzylidene)—2,4-dioxothiazolidin-3-yl)acetic acid (MBDTA-1) and (Z)—2-(5-(4-methoxybenzylidene)—2,4-dioxothiazolidin-3-yl)acetic acid (MBDTA-2).

Closer inspection of the crystal structure revealed the presence of four MBDTA-2 molecules bound at the centre of the homotetrameric protein (*Figure 5A*), in antiparallel pairs with each molecule, which were stabilised by interactions across three of the monomers (*Figure 5B*). Interestingly, this pocket, albeit distinct to the lysine binding site, shares two residues with it, namely W116 and

**Figure 4.** *In vitro* potency of DHDPS inhibitors. Dose responses of MBDTA-1 (• or ■) and MBDTA-2 (○ or □) against recombinant (**A**) AtDHDPS1 and (**B**) AtDHDPS2. Initial enzyme rates were normalised against a vehicle control (1% [v/v] DMSO). Normalised data (% activity remaining) is plotted as a function of $\log_{10}$[inhibitor] and fitted to a nonlinear regression model (solid line) ($R^2 = 0.99$). Data represents mean ± S.E.M. (N = 3).

The online version of this article includes the following source data for figure 4:

**Source data 1.** In vitropotency of DHDPS inhibitors.

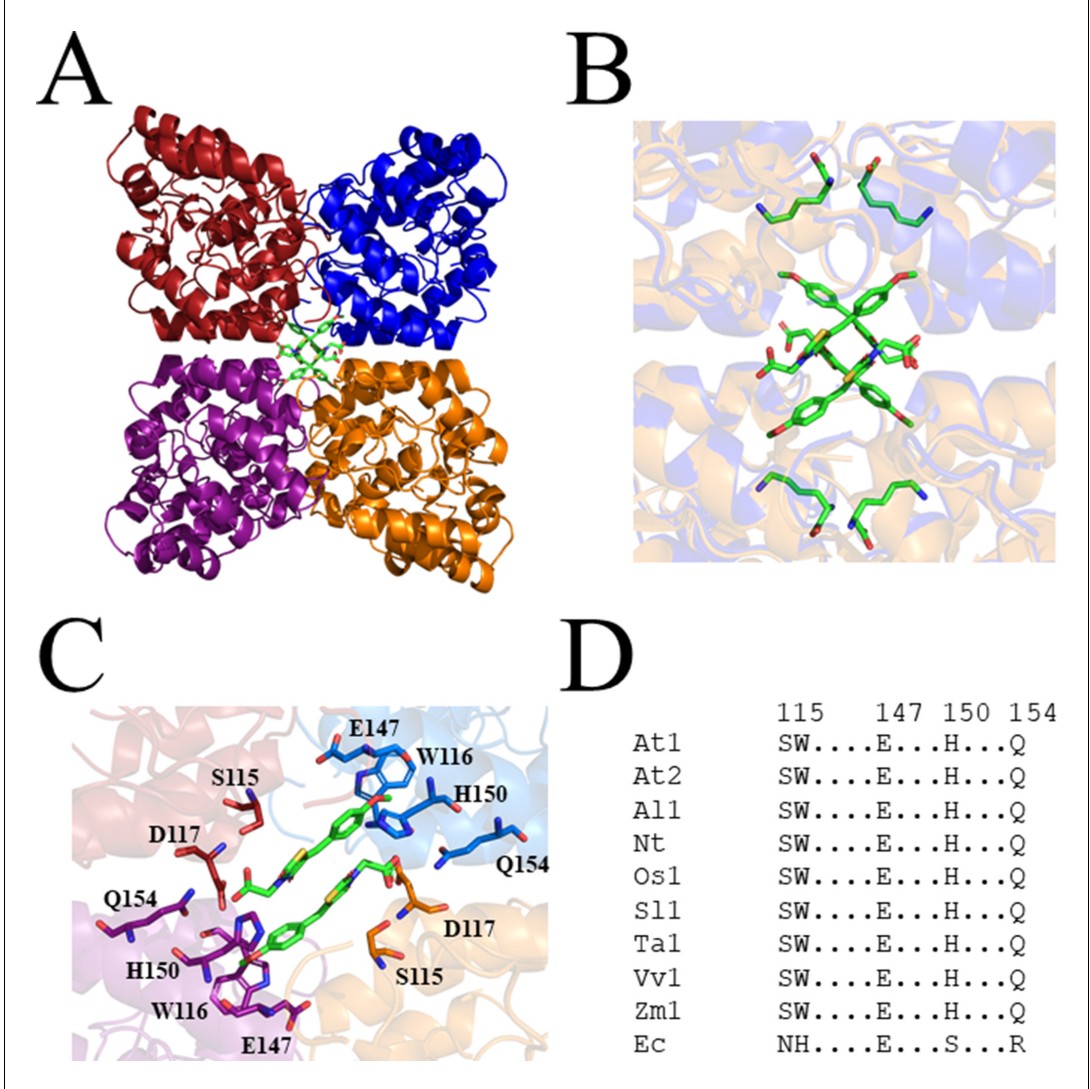

**Figure 5.** Crystal structure of AtDHDPS1 bound to MBDTA-2. (**A**) Cartoon view of overall AtDHDPS1 quaternary (tetrameric) structure, illustrating the binding sites for MBDTA-2 at the centre of the tetramer. (**B**) Overlay of the lysine-bound (PDB: 6VVH) and MBDTA-2-bound structures. (**C**) Close-up of inhibitor binding pocket, with interacting residues shown as sticks. Lysine and MBDTA-2 are shown as green sticks and coloured by nitrogen (blue), oxygen (red), and sulfur (yellow). Images were generated using *PyMOL* v 2.2 (Schrödinger). (**D**) Sequence alignment of residues involved in MBDTA-2 binding from *A. thaliana* DHDPS1 (At1; UNIPROT ID: Q9LZX6), *A. thaliana* DHDPS2 (At2; UNIPROT ID: Q9FVC8), *A. lyrata* DHDPS1 (Al1; UNIPROT ID: D7LRV3), *N. tabacum* DHDPS (Nt; UNIPROT ID: Q42948), *O. sativa* DHDPS1 (Os1; UNIPROT ID: A0A0K0K9A6), *S. lycopersicum* DHDPS1 (Sl1; UNIPROT ID: A0A3Q7IMG0), *T. aestivum* DHDPS1 (Ta1; UNIPROT ID: P24846), *V. vinifera* DHDPS1 (Vv1; UNIPROT ID: A0A438E022), *Z. mays* DHDPS1 (Zm1; UNIPROT ID: P26259), and *E. coli* (Ec) DHDPS (UNIPROT ID: P0A6L2). Residues are numbered according to *A. thaliana* DHDPS1 with dots (.) representing interspacing residues. Sequences were aligned in *BioEdit* (v 7.2.5) using the *ClustalW* algorithm.

The online version of this article includes the following figure supplement(s) for figure 5:

**Figure supplement 1.** Non-specific binding of MBDTA-2 between symmetry mates.

E147 (*Figure 2C*). Specifically, the methoxy group of MBDTA-2 interacts with W116, E147, and H150 from chain B, while the pendant carboxylic acid interacts with S115 from chain A as well as H150 and Q154 from chain C (*Figure 5C*). Additionally, we observed that upon binding, MBDTA-2 forces D117 to adopt a different rotamer conformation, which in turn, results in W116 assuming a different conformation. It must be noted that the four MBDTA-2 molecules were present with 50% occupancy. Consequently, each of the two moving residues, D117 and W116, adopt two distinct rotamer conformations, one of the apo- and ligand-bound states of AtDHDPS1. Given that no major rotamer changes or movement of catalytically important residues were noted, the exact mechanism of

**Table 1.** Summary of MBDTA-2-bound AtDHDPS1 crystallographic data collection, processing, and refinement statistics.

| Data collection | AtDHDPS1 + MBDTA-2 |
| --- | --- |
| Space group | P4$_1$2$_1$2 |
| Unit-cell parameters (Å) | 94.47, 94.47, 181.41 |
| Resolution (Å) | 20–2.29 (2.43–2.29) |
| No. of observations | 491,320 (74,297) |
| No. of unique reflections | 37,390 (5768) |
| Completeness (%) | 99.4 (96.6) |
| Redundancy | 13.1 (12.8) |
| $R_{merge}$ (%) | 9.9 (39.1) |
| $R_{meas}$ (%) | 10.0 (40.7) |
| CC$_{1/2}$ | 99.9 (97.8) |
| Average $I/\sigma(I)$ | 27.9 (7.9) |
| Refinement | |
| $R$ (%) | 18.3 |
| $R_{free}$ (%) | 22.6 |
| No. (%) of reflections in test set | 1071 |
| No. of protein molecules per asu | 2 |
| r.m.s.d bond length (Å) | 0.007 |
| r.m.s.d bond angle (°) | 1.415 |
| Average B-factors (Å$^2$)[*] | |
| Protein molecules | 44.52 |
| Ligand molecules | 60.01 |
| Water molecules | 40.33 |
| Ramachandran plot[†] | |
| Residues other than Gly and Pro in: | |
| Most favored regions (%) | 98.0 |
| Additionally allowed regions (%) | 2.0 |
| Disallowed regions (%) | 0.0 |
| PDB code | 7MDS |

Values in parentheses are for the highest-resolution shell.

[*]Calculated by *BAVERAGE* in CCP4 Suite (**Winn et al., 2011**).

[†]Calculated using *MolProbity* (**Chen et al., 2010**).

inhibition remains elusive. Nevertheless, this indicates the presence of a novel DHDPS allosteric pocket that has not been previously exploited for inhibitor discovery. Moreover, an alignment of the primary structure of several DHDPS enzymes from plant species indicates that the residues involved in MBDTA-2 binding are highly conserved across both monocotyledons and dicotyledons (*Figure 5D*) and therefore should allow for broad-spectrum inhibition.

## Specificity of DHDPS inhibitors

Following determination of the binding site, we examined the specificity of MBDTA-1 and MBDTA-2 to determine whether any future applications would have off-target effects. First, the cytotoxicity of the inhibitors was examined against the human cell lines, HepG2 and HEK293, using the 3-(4,5-dimethylthiazol-2-yl)−2,5-diphenyltetrazolium bromide (MTT) assay (*Figure 6A,B*). At the highest concentration assessed (400 µM), treatment with the inhibitors did not affect the viability of either cell line relative to the vehicle control. Second, the effect of the inhibitors on several bacterial species commonly found in the human flora and soil microbiome was assessed by measuring their minimum

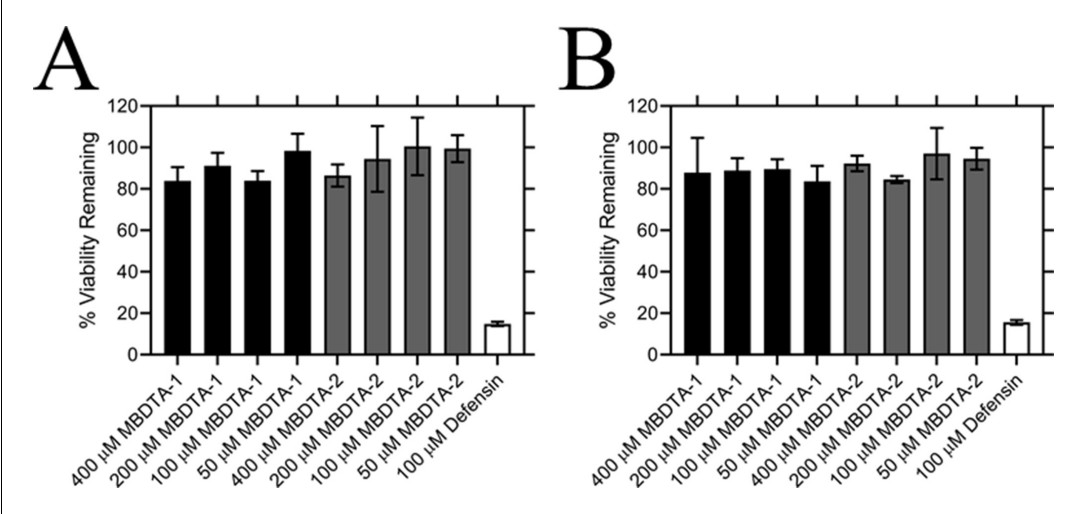

**Figure 6.** Effect of compounds on the viability of human cell lines. Toxicity of MBDTA-1 (black) and MBDTA-2 (grey), compared to the positive control defensin (white), assessed against (**A**) HepG2 and (**B**) HEK293 human cell lines using the MTT assay. Data were normalised against a vehicle control (1% [v/v] DMSO) and plotted against inhibitor concentration. Data represents mean ± S.E.M. (N = 3).

The online version of this article includes the following source data for figure 6:

**Source data 1.** Minimum inhibitory concentration(MIC) values for MBDTA-1 and MBDTA-2 against several bacterial strains.

inhibitory concentrations. No inhibition of bacterial growth was observed up to 128 µg·mL$^{-1}$ (equivalent to ~400 µM) (*Table 2*), indicating that these DHDPS inhibitors have specificity directed towards plants.

## Herbicidal efficacy

Given the promising *in vitro* properties of the inhibitors, we determined their herbicidal efficacy against *A. thaliana*, initially using seedling agar assays. At high micromolar concentrations of both MBDTA-1 and MBDTA-2, growth was completely attenuated, and most seeds were unable to germinate. Upon quantitation of root lengths, we determined an IC$_{50}$ of 98.1 ± 4.34 µM and 47.4 ± 0.450 µM for MBDTA-1 (*Figure 7A*) and MBDTA-2 (*Figure 7B*), respectively. Based on these results, we

**Table 2.** Minimum inhibitory concentration (MIC) values for MBDTA-1 and MBDTA-2 against several bacterial strains.

| | MBDTA-1 MIC (µg·mL$^{-1}$) | MBDTA-2 MIC (µg·mL$^{-1}$) |
|---|---|---|
| Human flora | | |
| *Enterococcus* spp. | >128 | >128 |
| *Staphylococcus aureus* | >128 | >128 |
| *Escherichia coli* | >128 | >128 |
| Soil bacteria | | |
| *Enterobacter ludwigii* | >128 | >128 |
| *Arthrobacter* sp. | >128 | >128 |
| *Enterobacter cancerogenus* | >128 | >128 |
| *Cedecea davisae* | >128 | >128 |
| *Rhodococcus erthropolis* | >128 | >128 |

The online version of this article includes the following source data for Table 2:

**Source data 1.** Minimum inhibitory concentration(MIC) values for MBDTA-1 and MBDTA-2 against several bacterial strains.

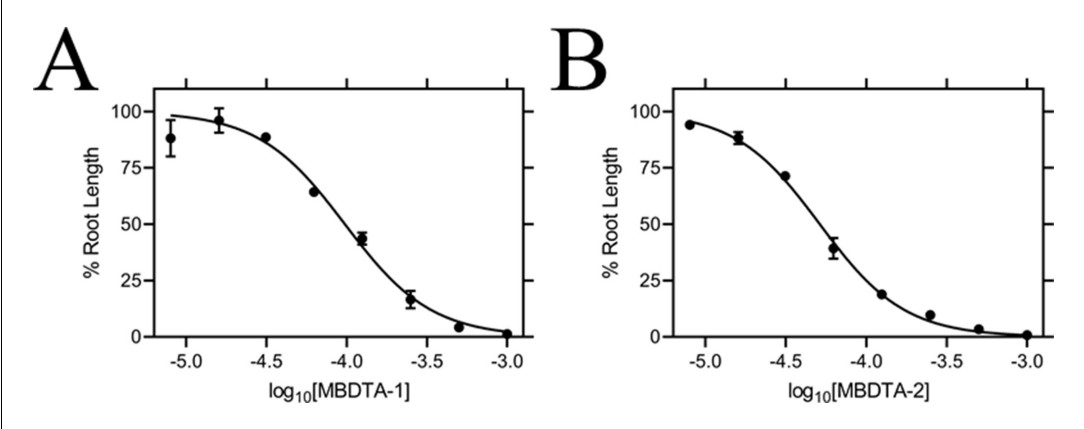

**Figure 7.** Effect of MBDTA compounds on agar-grown *A. thaliana*. A thaliana root lengths after treatment with increasing concentrations of either (**A**) MBDTA-1 or (**B**) MBDTA-2. Root lengths were determined using *ImageJ* v 1.53b and normalised against a vehicle control (1% [v/v] DMSO). Normalised data (•) (% root length) is plotted as a function of $\log_{10}$[inhibitor] and fitted to a nonlinear regression model (solid line) ($R^2 = 0.99$). Data represents mean ± S.E.M. (N = 3).

The online version of this article includes the following source data for figure 7:

**Source data 1.** Effect of MBDTA compounds on agar-grown *A. thaliana*.

examined their pre-emergence effect on soil-grown *A. thaliana*. Specifically, compounds were dissolved in a solution containing a non-ionic organic surfactant (Agral) and seeds were treated immediately after sowing on soil. The vehicle control-treated plants (*Figure 8A*) were used as a benchmark to visually assess the effects of inhibitors. The growth of *A. thaliana* in the presence of MBDTA-1 (*Figure 8B*) or MBDTA-2 (*Figure 8C*) at 300 mg·L$^{-1}$ was severely impeded as evidenced by the

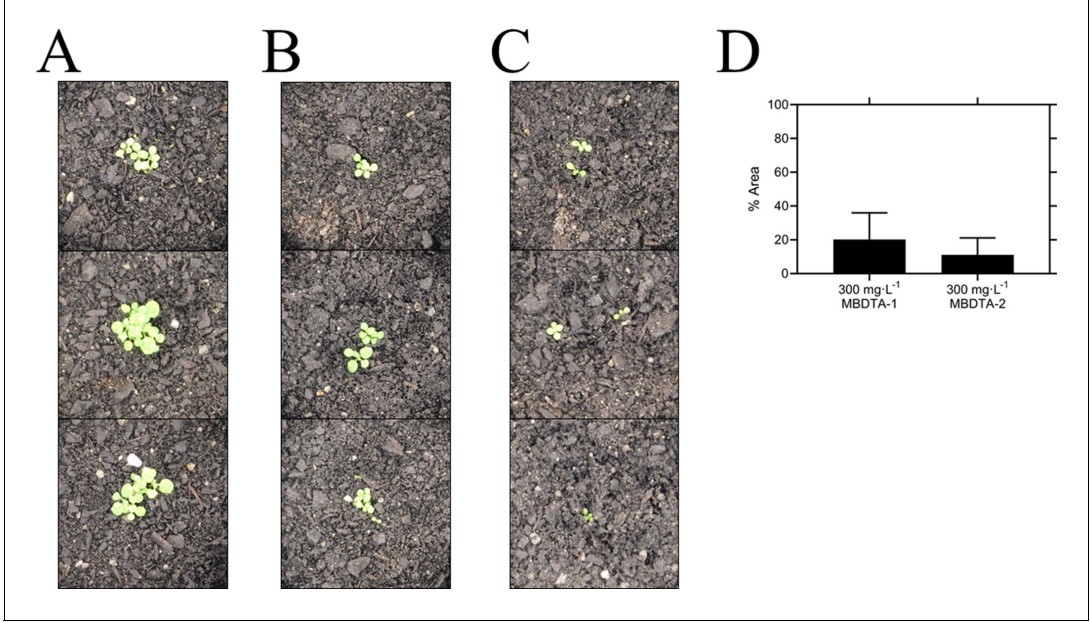

**Figure 8.** Pre-emergence efficacy of inhibitors on *A. thaliana* grown in soil. Treatments of (**A**) vehicle control (1% [v/v] DMSO), (**B**) 300 mg·L$^{-1}$ MBDTA-1, and (**C**) 300 mg·L$^{-1}$ MBDTA-2 given at day 0 (first day under controlled environment room conditions). A representative in triplicate of the biological replicates is shown vertically. (**D**) Leaf area of MBDTA-1/2-treated *A. thaliana*. Area was determined using *ImageJ* v 1.53b and normalised against a vehicle control (1% [v/v] DMSO). Data represents mean ± S.D. (n = 3).

The online version of this article includes the following source data for figure 8:

**Source data 1.** Pre-emergence efficacy of inhibitors on *A. thaliana* grown in soil.

growth area relative to the DMSO control (**Figure 8D**), wherein few seeds were able to germinate. This is consistent with the results observed at the highest concentrations of inhibitor on agar. Furthermore, the *A. thaliana* seeds capable of germinating in the presence of 300 mg·L$^{-1}$ MBDTA-2 were halted at the cotyledon stage before the generation of true leaves. As such, our newly discovered MBDTA compounds represent the first DAP pathway inhibitors with soil efficacy against plants.

## Discussion

The lack of herbicides with novel modes of action entering the market in the past three decades has led to an over-reliance on our current agrichemicals, which has contributed to the rapid generation of resistance. Although the DAP pathway has gained attention as a way to increase the nutritional content of lysine in crops (*Wang et al., 2017*), it has remained an unexplored target for the development of herbicides until now. DHDPS catalyses the first step of the DAP pathway and is commonly duplicated in plant species, including *A. thaliana*. Both DHDPS proteins are localised to the chloroplast and share >85% of primary structure identity, with the majority of differences at the N-terminus. Although DHDPS is essential in *A. thaliana* (*Jones-Held et al., 2012*), there have been no published inhibitors of the plant enzymes, with much of the focus on inhibitors of bacterial DHDPS as possible new antibiotics. Several iterations of active site inhibitors against bacterial DHDPS enzymes have been developed, but most of these have, at best, high micromolar potency (*Christoff et al., 2019*). As the active site of DHDPS is seemingly difficult to target, the allosteric site of the enzyme could represent a more fruitful route for the development of inhibitors. Indeed, lysine is the most potent inhibitor of plant DHDPS orthologues. Furthermore, the most potent bacterial DHDPS inhibitor to date is bislysine, whereby two lysine molecules are linked together via an ethylene bridge (*Skovpen et al., 2016*). Although herbicides commonly directly target, or bind closely to, the active site of their respective enzyme target, this study shows that allosteric site inhibitors can be employed and afford lethal phytotoxicity.

Our study describes the discovery of the first plant DHDPS inhibitors, with two MBDTA analogues identified and characterised here. The mode of inhibition is via a novel binding pocket adjacent to the lysine binding site, which results in the allosteric inhibition of the enzyme. Lysine has recently been shown to differentially inhibit the AtDHDPS isoforms, with AtDHDPS1 being 10-fold more sensitive to the allosteric inhibitor (*Hall et al., 2021*). In this study, we demonstrate that the MBDTA compounds have similar inhibitory effects against both enzymes. This further supports crystallography data demonstrating that MBDTA-2 binds in a pocket adjacent to the lysine allosteric site and is likely acting in a different way. However, the exact mechanism of inhibition, much like lysine-mediated allostery, remains elusive. Moreover, this binding pocket is conserved across multiple plant species, including both monocots and dicots. Importantly, our compounds lacked off-target toxicity, whilst resulting in the inhibition of germination and growth of *A. thaliana* seedlings on solidified media and in soil. However, as expected, plant inhibition was more pronounced on media likely due to the stability, distribution, and persistence of the compounds. Nevertheless, the assays performed on soil demonstrate their potential applicability as pre-emergence treatments. It would also be of interest to investigate the metabolic shifts in plants treated with inhibitors and determine if there is a toxic build-up of other amino acids such as threonine, which has been observed in *DHDPS* knockout experiments (*Sarrobert et al., 2000*). Indeed, a common trait of systemic herbicides is that their efficacy is often related to the cascading consequences of inhibiting a key reaction, rather than inhibition of the reaction itself (*Hall et al., 2020*). Additionally, it would be of interest to determine the metabolic state of the compounds after uptake into the cells. This would elucidate whether the MBDTA compounds act as proherbicides, potentially through the demethylation of their aryl methyl ethers. As such, it would also be of interest to test demethylated analogues *in vitro* and *in planta* to assess any changes in potency relative to the methoxy compounds.

Developing enzyme inhibitors into a commercial product is an arduous and costly process. Optimisation of phytotoxicity, water solubility, cell wall penetration, translocation, soil/water persistence, and formulation must all be considered. The MBDTA compounds described here represent an attractive avenue to pursue, and with the elucidation of a novel binding pocket within DHDPS, it may be possible to rationally improve their potency guided by the crystallography data. Specifically, it would be of interest to build compounds that extend into the lysine binding pocket, which could improve potency. Alternatively, novel chemical scaffolds could be explored to target the DHDPS

pocket identified. The inhibitors must be able to traverse the chloroplast membrane in order to reach the DHDPS target and be amenable to post-emergence application. It would also be of interest to study inhibitors with increased hydrophobicity and thus, potentially enhanced transport through the epidermis, cell wall, and plastid membrane, to reach the DHDPS target. However, it is important to recognise that there is a fine balance between membrane permeability and aqueous solubility that must be achieved for any new herbicides. Following the synthesis of more potent analogues, it would be of interest to test inhibitors against weed species. To further pursue the compounds described herein and eventually deploy more potent analogues as herbicides, tolerant crops will eventually need to be engineered. As the MBDTA binding pocket is highly conserved among plant species, these inhibitors will likely have broad-spectrum phytotoxicity. Given that the inhibitors do not bind at the active site, it should be possible to engineer tolerant crops with mutant DHDPS enzymes that are not susceptible to inhibition with minimal effects on enzyme function. Importantly, DHDPS inhibitors could also be used in conjunction with other herbicides as part of a combinatorial treatment to yield synergistic responses and circumvent resistance mechanisms to tackle the global rise in herbicide-resistant weeds.

# Materials and methods

**Key resources table**

| Reagent type (species) or resource | Designation | Source or reference | Identifiers | Additional information |
|---|---|---|---|---|
| Gene (*Arabidopsis thaliana*) | *DHDPS1* | TAIR | *At3G60880* | |
| Gene (*Arabidopsis thaliana*) | *DHDPS2* | TAIR | *At2G45440* | |
| Cell line (*Homo sapiens*) | HepG2 | ATCC | ATCC: HB-8065 RRID:CVCL_0027 | |
| Cell line (*Homo sapiens*) | HEK293 | ATCC | ATCC:ACS-4500 RRID:CVCL_0063 | |

## High-throughput chemical screen and analogue synthesis

A high-throughput screen of a library of 87,648 compounds was conducted against recombinant DHDPS enzyme by the Walter and Eliza Hall Institute High Throughput Chemical Screening Facility (Melbourne, Australia). The *o*-ABA colourimetric assay employed assesses DHDPS activity via the formation of a purple chromophore that can be measured at 520–540 nm (*Yugari and Gilvarg, 1965*). The assay was miniaturised, so it could be performed in 384-well plates. For the primary screen, reactions comprised 0.5 mg·mL$^{-1}$ DHDPS, 0.5 mM sodium pyruvate, and 0.5 mM ASA. Library compounds were added at final concentrations of 20 mM, with DMSO concentrations kept at 0.4% (v/v). After ASA addition, reactions were incubated at 25℃ for 15 min, before a final concentration of 350 mM HCl was added to stop the reaction. *o*-ABA was subsequently added to a final concentration of 0.44 mg·mL$^{-1}$, plates incubated at room temperature for 1 hr, and absorbance quantified at 540 nm. Vehicle (DMSO) was used as positive controls, and negative controls lacked ASA. For the secondary screen, 11-point dose–response curves were generated using the same reactions as described above. A counter screen was conducted using the same set-up albeit without the inclusion library compounds before the addition of 350 mM HCl. Library compounds were then added after the reaction was stopped, followed by *o*-ABA to a final concentration of 0.44 mg·mL$^{-1}$. The plates were subsequently incubated at room temperature for 1 hr, and absorbance was quantified at 540 nm. Analogues were designed and synthesised using the methods described in previous and contemporary work (*Perugini et al., 2018*).

## Expression and purification of *A. thaliana* DHDPS enzymes

Both DHDPS isoforms from *A. thaliana* were expressed and purified as previously described (*Hall et al., 2021*). Briefly, AtDHDPS isoforms were expressed in *Escherichia coli* BL21 (DE3) cells, with AtDHDPS2 requiring the GroEL/ES chaperone complex to facilitate correct folding. Purification was performed using immobilised metal affinity chromatography. Lastly, fusion tags were cleaved by

human rhinovirus 3C or tobacco etch virus protease for AtDHDPS1 and AtDHDPS2, respectively, whilst simultaneously dialysing into storage buffer (20 mM Tris, 150 mM NaCl, 0.5 mM TCEP, pH 8.0).

## Enzyme kinetics

DHDPS enzyme activity was determined using the DHDPS–DHDPR coupled assay as previously described by measuring the oxidation of NADPH (*Atkinson et al., 2013*; *Hall et al., 2021*). Assays were carried out in a Cary 4000 UV/Vis spectrophotometer at 30°C with substrates fixed at the previously determined Michaelis–Menten constant values (*Griffin et al., 2012*; *Hall et al., 2021*). Inhibitor was titrated against AtDHDPS enzymes, and reactions were incubated at 30°C for 12 min before initiation with ASA. Initial velocity data were normalised against a vehicle (DMSO) control and analysed using *Equation 1* (log(inhibitor) vs. normalised response – variable slope, *GraphPad Prism* v 8.3). Dose responses were performed with three technical replicates for each concentration of compound. Dose responses were repeated with three biological replicates, each using a new stock of reagents.

$$Y = 100/(1 + 10^{((\text{LogIC}_{50} - X) \times \text{HillSlope})}) \tag{1}$$

where Y is the normalised rate, $\text{logIC}_{50}$ is the logarithmic concentration of ligand resulting in 50% activity, X is the concentration of ligand, and Hill Slope is the steepness of the curve.

## X-ray crystallography

AtDHDPS1 was co-crystallised as previously described in the presence of MBDTA-2 (*Hall et al., 2021*). Briefly, protein (8.5 mg·mL$^{-1}$) was incubated at 20°C with MBDTA-2 at a final concentration of 1 mM (in 2% [v/v] DMSO) before being added in a 1:1 ratio to a reservoir solution containing 1.4 M $(NH_4)_2SO_4$, 0.1 M NaCl, 0.1 M HEPES (pH 7.5), and 1 mM MBDTA-2 (in 2% [v/v] DMSO). Plates were incubated at 20°C. Crystals were briefly dipped in cryo-protectant (1.4 M $(NH_4)_2SO_4$, 0.1 M NaCl, 0.1 M HEPES [pH 7.5], 1 mM MBDTA-2 [in 2% (v/v) DMSO], and 20% [v/v] glycerol) and flash frozen in liquid nitrogen. Data were collected at the Australian Synchrotron using the MX2 beamline (*Aragão et al., 2018*). A total of 1800 diffraction images were collected with 0.1° oscillation using an EIGER 16M detector at a distance of 350 mm, with 20% beam attenuation for a total exposure time of 18 s. X-ray data were integrated using *XDS* (*Kabsch, 2010*) and scaled with *AIMLESS* (*Evans and Murshudov, 2013*) before phases were determined by molecular replacement through *Auto-Rickshaw* (*Panjikar et al., 2005*) with AtDHDPS1 (PDB ID: 6VVI) used as a search model (*Hall et al., 2021*). Manual building was performed in *COOT Emsley et al., 2010* followed by refinement employing *REFMAC5* in the *CCP4i2* (v7.0) software suite (*Emsley et al., 2010*; *Murshudov et al., 2011*; *Winn et al., 2011*). SMILES string of the inhibitor (MBDTA-2) was processed through *AceDRG* to generate the coordinate and cif file (*Long et al., 2017*). Validation was completed using *MolProbity* (*Chen et al., 2010*). The structure of MBDTA-2 bound to AtDHDPS1 is deposited in the Protein Data Bank as 7MDS.

## Cell lines

Cell lines used were sourced from the American Type Culture Collection and were authenticated using STR DNA profiling, and no mycoplasma contamination was detected.

## Toxicity assays

The toxicity of inhibitors against human HepG2 and HEK293 cell lines was assessed using the MTT viability assay as previously described (*Soares da Costa et al., 2012*). In brief, the cells were suspended in Dulbecco-modified Eagle's medium containing 10% (v/v) fetal bovine serum and then seeded in 96-well tissue culture plates at 5000 cells per well. After 24 hr, cells were treated with 50–400 µM of MBDTA-1 or MBDTA-2, such that the DMSO concentration was consistent at 1% (v/v) in all wells. Alternatively, cells were treated with the cytotoxic defensin protein at 100 µM (*Baxter et al., 2017*). After treatment for 48 hr, MTT cell proliferation reagent was added to each well and incubated for 3 hr at 37°C. The percentage viability remaining reported is relative to the vehicle control of 1% (v/v) DMSO. Assays were performed in three biological replicates, using a different batch of reagents and cells.

## Antibacterial assays

The minimum inhibitory concentration (MIC) for MBDTA-1 and MBDTA-2 was determined against a panel of Gram-positive and Gram-negative bacteria using a broth microdilution method according to guidelines defined by the Clinical Laboratory Standards Institute (*National Committee for Clinical Laboratory Standards, 2004*; *National Committee for Clinical Laboratory Standards, 2003*). An inoculum of $1 \times 10^5$ colony forming units/mL was used, and the testing conducted using tryptic soy broth in 96-well plates. Growth was assessed after incubation at 37°C for 20 hr by measuring the absorbance at 600 nm. The MIC value is defined as the lowest concentration of inhibitor where no bacterial growth is observed. Experiments were performed in three biological replicates, using a different stock of reagents and bacterial culture.

## Seedling assays

Inhibitors were dissolved in 1× Gamborg modified/Murashige Skoog (GM/MS) media to final concentrations of 8–1000 µM. Specifically, media were prepared with 0.8% (w/v) plant grade agar and 1% (w/v) sucrose before sterilisation (*Lindsey et al., 2017*). *A. thaliana* seeds were surface sterilised by soaking in 80% (v/v) ethanol for 5 min, followed by a 15 min incubation in bleach solution containing 1% (v/v) active NaClO and rinsed in excess sterile water before placing onto agar-containing inhibitors (*Boyes et al., 2001*). *A. thaliana* seeds were stratified at 4°C for 72 hr in the dark prior to relocation to a controlled environment room (CER), where seeds were grown at 22 ± 0.5°C at 60 ± 10% humidity with light produced by cool white fluorescent lights at a rate of ~110 µmol·m$^{-2}$·s$^{-1}$ over long-day conditions (16 hr light:8 hr dark) (*Boyes et al., 2001*). Plates were positioned upright to allow roots to grow downwards, and after 7 days, images were taken, and root length was determined using *ImageJ* (v 1.53b) (*Rasband, 2011*). Outliers were identified using the 1.5× interquartile range method (*Tukey, 1977*). Resulting data were analysed using *Equation 1* (log(inhibitor) vs. normalised response – variable slope, *GraphPad Prism* v 8.3). No DMSO and vehicle (1% [v/v] DMSO) controls were also employed. Assays were carried out with 20 technical replicates (i.e. seeds) per experiment and were repeated in three biological replicates, with each biological replicate using a different stock of reagents and batch of seeds.

## Soil assays

Inhibitors were prepared in DMSO and diluted to 300 mg·L$^{-1}$ (1% [v/v] DMSO) in H$_2$O containing 0.01% (v/v) Agral (Syngenta, North Ryde, NSW, Australia). *A. thaliana* seeds were surface sterilised as above and resuspended in 0.1% (w/v) agar before stratification. Subsequently, ~30 seeds were sown into a small indentation (depth ~1 cm) in moist seed raising soil (pH 5.5) (Biogro, Dandenong South, VIC, Australia), supplemented with 0.22% (w/w) Nutricote N12 Micro 140 day-controlled release fertiliser (Yates, Sydney, NSW, Australia). Seeds were treated with 1 mL (equivalent to 1200 g·ha$^{-1}$) of compound or vehicle control by pipetting and covered with soil just prior to transfer to a CER and images taken after 7 days. Area analysis was performed using colour thresholding in *ImageJ* (v 1.53b) and normalised against the DMSO control (*Corral et al., 2017*; *Rasband, 2011*). Assays were carried out across three technical replicates (i.e. pots) using the same batch of reagents and seed stock.

## Acknowledgements

TPSC would like to thank the National Health and Medical Research Council of Australia (APP1091976) and Australian Research Council (DE190100806) for fellowship and funding support, and MAP and SP thank the Australian Research Council for funding support (DP150103313). ARG would like to thank the Australian Research Council Research Hub for Medicinal Agriculture (IH180100006) for support. CJH is supported by La Trobe University Postgraduate Research scholarships. RMC is a recipient of an Australian Government Research Training Program Scholarship and a LIMS Write-Up Award. This research was supported by the Defence Science Institute, an initiative of the State Government of Victoria, with a scholarship awarded to JAW, who is also the recipient of a Research Training Program scholarship. We thank Dr Grant Pearce (University of Canterbury, New Zealand) for supplying pET151/D-Topo harbouring the DHDPS2/*dapA2* gene and Professor Ashley Franks (La Trobe University, Australia) for supplying soil bacterial isolates. We acknowledge the use

of the MX2 beamline at the Australian Synchrotron, part of ANSTO and employed the Australian Cancer Research Foundation (ACRF) detector. We acknowledge the CSIRO Collaborative Crystallisation Centre (http://www.csiro/C3; Melbourne, Australia). We also thank the La Trobe University Comprehensive Proteomics Platform for providing infrastructure support.

# Additional information

## Competing interests

Tatiana P Soares da Costa, Belinda M Abbott, Matthew A Perugini: is listed as an inventor on a patent pertaining to inhibitors described in the manuscript. Patent Title: Heterocyclic inhibitors of lysine biosynthesis via the diaminopimelate pathway; International patent (PCT) No.: WO2018187845A1; Granted: 18/10/2018. The other authors declare that no competing interests exist.

## Funding

| Funder | Grant reference number | Author |
| --- | --- | --- |
| National Health and Medical Research Council of Australia | APP1091976 | Tatiana P Soares da Costa |
| Australian Research Council | DE190100806 | Tatiana P Soares da Costa |
| Australian Research Council | DP150103313 | Santosh Panjikar<br>Matthew A Perugini |
| Australian Research Council | IH180100006 | Anthony R Gendall |

The funders had no role in study design, data collection and interpretation, or the decision to submit the work for publication.

## Author contributions

Tatiana P Soares da Costa, Conceptualization, Resources, Data curation, Formal analysis, Supervision, Funding acquisition, Validation, Investigation, Visualization, Methodology, Writing - original draft, Project administration, Writing - review and editing; Cody J Hall, Data curation, Formal analysis, Validation, Investigation, Visualization, Methodology, Writing - original draft, Writing - review and editing; Santosh Panjikar, Resources, Formal analysis, Validation, Investigation, Methodology, Writing - review and editing; Jessica A Wyllie, Formal analysis, Validation, Investigation, Methodology, Writing - review and editing; Rebecca M Christoff, Saadi Bayat, Resources, Methodology, Writing - review and editing; Mark D Hulett, Resources, Writing - review and editing; Belinda M Abbott, Resources, Supervision, Visualization, Methodology, Writing - review and editing; Anthony R Gendall, Resources, Supervision, Funding acquisition, Methodology, Writing - review and editing; Matthew A Perugini, Conceptualization, Resources, Supervision, Funding acquisition, Methodology, Project administration, Writing - review and editing

## Author ORCIDs

Tatiana P Soares da Costa https://orcid.org/0000-0002-6275-7485
Santosh Panjikar http://orcid.org/0000-0001-7429-3879
Mark D Hulett http://orcid.org/0000-0003-2072-5968
Anthony R Gendall http://orcid.org/0000-0002-2255-3939

## Decision letter and Author response

Decision letter https://doi.org/10.7554/eLife.69444.sa1
Author response https://doi.org/10.7554/eLife.69444.sa2

# Additional files

## Supplementary files

- Supplementary file 1. Physicochemical properties of MBDTA-1/2.

• Transparent reporting form

### Data availability

Diffraction data have been deposited in PDB under the accession code 7MDS. The validation report has been uploaded as a 'Related Manuscript File'. Other data sets have been uploaded as 'Source Data' files.

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
