## [Decision Letter]

**Acceptance summary:**

This manuscript reports two compounds identified out of a large library screen that have inhibition activity on dihydrodipicolinate synthase (DHDPS), a two copy gene in most plants that is essential in Arabidopsis and is the rate-limiting step for lysine synthesis. Inhibition was shown using an in vitro assay on recombinant DHDPS, a crystal model was used to identify possible binding sites, and biological data in the form of plant growth inhibition on agar and in soil were reported to demonstrate proof of concept for herbicidal activity. Lysine biosynthesis inhibition has high potential as an important new herbicide mode of action to address the global issue of multiple herbicide-resistant weeds.

**Decision letter after peer review:**

Thank you for submitting your article "Towards Novel Herbicide Modes of Action by Inhibiting Lysine Biosynthesis in Plants" for consideration by *eLife*. Your article has been reviewed by 3 peer reviewers, including Todd Gaines as the Reviewing Editor and Reviewer #2, and the evaluation has been overseen Detlef Weigel as the Senior Editor. The following individuals involved in review of your submission have agreed to reveal their identity: Stephen Duke (Reviewer #1); Franck Dayan (Reviewer #3).

Essential revisions:

1) Reviewer #1 noted that the I50 values for inhibition of the putative target enzyme and for inhibition of Arabidopsis growth on agar were similar, which suggests that the chemical structures (both methoxy compounds) of the two herbicide candidates may be proherbicides, possibly demethoxylated in vivo to compounds that are more active at the target site. The authors are requested to revise the discussion to explore the possibility that the compounds may be proherbicides, and describe follow-up experiments that would be needed to test this hypothesis.

2) Reviewer #3 noted that the tested compounds may be allosteric inhibitors; the authors are requested to revise the manuscript to address this point and possible implications for developing commercial inhibitors of this enzyme.

*Reviewer #1:*

The authors provide candidates for a new herbicide mode of action – something that is badly needed. This well-written paper provides good data at the enzyme and whole plant level. However, the similar I50 values at these two levels is surprising. I think that the two compounds are probably proherbicides. More data will be needed to determine this.

These authors provide data in support of two new compounds being herbicidal as a result of inhibition of the first enzyme of the lysine biosynthesis pathway – a new herbicide mode of action. The paper is well written and good as far as it goes, but the similar I50 values for inhibition of the putative target enzyme and for inhibition of arabidopsis growth on agars are surprising, as target enzyme I50 values for herbicides are normally much lower than whole plant herbicide I50 values. The chemical structures (both methoxy compounds) of the two herbicide candidates indicates that they may be proherbicides – that is, they are probably demethoxylated in vivo to compounds that are more active at the target site. Nevertheless, this paper adds significantly to what is known of herbicides that act on amino acid synthesis and will be of great interest to those who work in this field.

Any information about a potential new mode of action for herbicides is welcome. This is a well-written paper. However, there is a major issue that should be easily sorted out with a couple of simple experiments.

This issue is that the in vitro activity of the two compounds on the enzyme is similar to the in vivo activity on the Arabidopsis on agar. Usually, the in vitro I50 value for the enzyme is much lower than that for as a herbicide on whole plants. Looking at the structures of the two compounds, I would guess that both of them are demethylated in vivo (like diclofop-methyl). If so, the molecules that reach the target are the demethylated versions of these compounds. If I am right, the I50 values of the demethylated compounds on the enzyme will be much lower that what is reported for the O-methylated forms, and the I50 values of the demethylated compounds as herbicides on Arabidopsis in agar will be much higher than that of the O-methylated forms. These experiments should be done in future work, and this possibility should be discussed in the paper because it will be obvious to most scientists who deal with herbicide chemistry and mode of action.

*Reviewer #2:*

This manuscript reports convincing data showing several compounds identified out of a large library screen that have inhibition activity on dihydrodipicolinate synthase (DHDPS), a two copy gene in most plants previously shown to be essential in Arabidopsis and previously shown by this research group to be the rate limiting step for lysine synthesis. Inhibition was shown using an in vitro assay on recombinant DHDPS, a crystal model was used to identify possible binding sites, and biological data in the form of plant growth inhibition on agar and in soil were reported to demonstrate herbicidal activity. Lysine biosynthesis inhibition has high potential as an important new herbicide mode of action.

Strengths:

The combination of biochemical, crystallography, and biological data was very convincing to demonstrate the herbicidal activity of the tested compounds for lysine synthesis inhibition. The paper is novel in that while DHDPS has previously been shown to be essential in plants (double knockouts of the two genes are lethal), this is the first report of chemical inhibitors of DHDPS in plants. This would represent a new mode of action for herbicides, which is a major finding. The identified inhibitors have biological activity at reasonable concentrations that could be applicable in agriculture. The claims made by the authors are justified based on their data. All experiments are technically sound, the results are clearly presented, and the supporting data are all fully available.

Weaknesses:

Some questions remain about the potential utility of the identified inhibitors as effective herbicides. The soil application experiment showed biological activity by reducing plant growth, but the possible use of the inhibitors for post-emergence weed control was not explored. The inhibitors would likely be non-selective, meaning they would also damage crops, so the potential for generating resistant crops by identifying metabolic detoxification genes from bacteria could be explored.

Comments for the authors:

This is an outstanding manuscript. I have a few comments for some points that could be addressed to further improve the paper.

Regarding the soil experiment, was the compound sprayed onto the soil before placing the seeds, or was the compound sprayed directly on the seeds? Were seeds covered by soil, whether before or after spraying, and if covered, to what depth of soil? This is important to know if the seedlings are absorbing the herbicide from the soil-water solution, or if they were contacted directly by the spray droplets at application. A few more details needed on the soil, such as composition (pH; percent sand, silt, and clay; percent organic matter). Regarding the application of the herbicide solution to the soil, the authors state that 1 mL was applied, but how? Was it pipetted on or sprayed? The 1 mL volume is quite a lot for the small area, and may represent an unrealistically high quantity. Also need to state volume of solution sprayed on soil (the solution was 300 mg/L of the herbicide but what volume of this was applied to the soil?) This is important to enable replication and testing by others.

The authors state that there are two copies of the DHDPS gene in plants; suggest that they consider reporting published Arabidopsis RNA seq data for these genes to state what is their relative expression level, whether relatively low, mid, or high? And is there any tissue or developmental specificity known for these genes? These questions are relevant because an ideal herbicide target has lower expression, thereby making it easier to inhibit.

This herbicide would apparently be non-selective in crops, given that it has conserved sequence in rice and Arabidopsis, grape, wheat. The authors could perhaps explore a few more dicot plant families containing important weeds to determine if sequence conservation is maintained. Also there is the question of how this herbicide would be used in crops; if non-selective, a resistance transgene would be needed. The authors could consider proposing in the discussion to run bacteria screens to look for metabolism genes to engineer crops?

Regarding how the herbicide would be used, what about leaf uptake? What are the physicochemical properties of the compounds? Predictions about binding to soil? Consider running them through a predictor to report log Kow, any pKa that would be relevant for uptake and acid trapping, and try to predict if these would translocate or not. A potentially helpful reference for this would be:

Gandy, M.N., Corral, M.G., Mylne, J.S. and Stubbs, K.A., 2015. An interactive database to explore herbicide physicochemical properties. Organic & biomolecular chemistry, 13(20), pp.5586-5590.

Line 345-346: hydrophilicity is not generally expected to increase movement across cell wall, should this be hydrophobicity? Again the log Kow to indicate water solubility should be reported. In addition to the cell wall as a barrier, herbicides must pass the cell membrane and if plastid targeted, the plastid membrane, along with passing the wax and cuticle on the epidermis; consider revising this discussion point to expand that the cell wall is not the only barrier for uptake.

*Reviewer #3:*

The discovery of herbicide target sites is a difficult endeavor. While there are hundreds of targets that can be inhibited, it turns out that very few of them provide suitable lethality to develop commercial herbicides. This study investigates dihydrodipicolinate synthase (the rate limiting step in lysine biosynthesis as a potential herbicide target site) and identifies a couple of molecules that provide some level of inhibition. Binding of these inhibitors is characterized biochemically and via crystallography.

Lysine biosynthesis is an essential amino acid, so only plants and microbes have the capacity to synthesize it. This makes it a particularly attractive pathway to develop herbicides because animals (humans) don't have that pathway. So the rational for the study is valid.

This study investigates dihydrodipicolinate synthase (the rate limiting step in lysine biosynthesis as a potential herbicide target site) and identifies a couple of molecules that provide some level of inhibition. Generally speaking this study is done properly and the experiments appear to have been done correctly. I have made several grammatical/typographical suggestions directly on the manuscript.

First, the authors do a good job demonstrating that DHDPS can be inhibited, establishing the fact that it might be a suitable target site. The two most active molecules are not very potent, so I am glad the authors don't oversell the potency of the molecules they discovered. However, it is interesting that these compounds have some soil activity, albeit at very high concentrations.

I find the portion of the manuscript characterizing the binding of these inhibitors could be developed a little more.

1. At first glance, I am not sure if the pose of the inhibitors shown in Figure 5 is a true reflection of their binding on DHDPS. These poses may be a crystallization artifact. The inhibitors appear to interact with each other via pi pi stacking and I wonder if they would crystalize like this if by themselves.

2. The authors did note that some histidine residues moved to accommodate for the presence of the inhibitors. However, one would expect that the inhibitors binding would be stabilized by pi pi stacking with the histidine or the tryptophan residues within the proximity of the molecules, but this does not appear to be the case. May be the authors should discuss this.

3. The binding of these inhibitors appears to overlap with the allosteric binding site of lysine. This may explain why the potency of the inhibitors is not very high. To my knowledge, no commercial herbicide is known to target an allosteric site. Their may be come biochemical reasons for this. Allosteric regulation by inhibitors may not be 'lethal' enough. The authors should discuss this possibility. I suggest that they also include a statement indicating that other inhibitors interacting with the catalytic domain of DHDPS might lead to more active herbicide leads.

Nonetheless, this article is very interesting and provides new information.

The authors will not have any problem addressing the grammatical/typographical suggestions I made directly on the manuscript (see attached PDF).

---

## [Author Response]

Essential revisions:1) Reviewer #1 noted that the I50 values for inhibition of the putative target enzyme and for inhibition of Arabidopsis growth on agar were similar, which suggests that the chemical structures (both methoxy compounds) of the two herbicide candidates may be proherbicides, possibly demethoxylated in vivo to compounds that are more active at the target site. The authors are requested to revise the discussion to explore the possibility that the compounds may be proherbicides, and describe follow-up experiments that would be needed to test this hypothesis.

We thank the reviewer for highlighting this important point. We have addressed this by adding the following to the Discussion:

Lines 355-360: “Additionally, it would be of interest to determine the metabolic state of the compounds after uptake into the cells. […] As such, it would also be of interest to test demethylated analogues in vitro and in planta to assess any changes in potency relative to the methoxy compounds.”

2) Reviewer #3 noted that the tested compounds may be allosteric inhibitors; the authors are requested to revise the manuscript to address this point and possible implications for developing commercial inhibitors of this enzyme.

We thank the reviewer for the comment. We have revised the manuscript to address this point and possible implications. The following has been added to the Discussion:

Lines 325-334: “Several iterations of active site inhibitors against bacterial DHDPS enzymes have been developed, but most of these have, at best, high micromolar potency (Christoff et al., 2019). As the active site of DHDPS is seemingly difficult to target, the allosteric site of the enzyme could represent a more fruitful route for the development of inhibitors. Indeed, lysine is the most potent inhibitor of plant DHDPS orthologues. Furthermore, the most potent bacterial DHDPS inhibitor to date is bislysine, whereby two lysine molecules are linked together via an ethylene bridge (Skovpen et al., 2016). Although herbicides commonly directly target, or bind closely to, the active site of their respective enzyme target, this study shows allosteric site inhibitors can be employed and afford lethal phytotoxicity.”

Lines 379-382: **“**Given that the inhibitors do not bind at the active site, it should be possible to engineer tolerant crops with mutant DHDPS enzymes that are not susceptible to inhibition with minimal effects on enzyme function.”

Reviewer #1:The authors provide candidates for a new herbicide mode of action – something that is badly needed. This well-written paper provides good data at the enzyme and whole plant level. However, the similar I50 values at these two levels is surprising. I think that the two compounds are probably proherbicides. More data will be needed to determine this.These authors provide data in support of two new compounds being herbicidal as a result of inhibition of the first enzyme of the lysine biosynthesis pathway – a new herbicide mode of action. The paper is well written and good as far as it goes, but the similar I50 values for inhibition of the putative target enzyme and for inhibition of arabidopsis growth on agars are surprising, as target enzyme I50 values for herbicides are normally much lower than whole plant herbicide I50 values. The chemical structures (both methoxy compounds) of the two herbicide candidates indicates that they may be proherbicides – that is, they are probably demethoxylated in vivo to compounds that are more active at the target site. Nevertheless, this paper adds significantly to what is known of herbicides that act on amino acid synthesis and will be of great interest to those who work in this field.Any information about a potential new mode of action for herbicides is welcome. This is a well-written paper. However, there is a major issue that should be easily sorted out with a couple of simple experiments.This issue is that the in vitro activity of the two compounds on the enzyme is similar to the in vivo activity on the Arabidopsis on agar. Usually, the in vitro I50 value for the enzyme is much lower than that for as a herbicide on whole plants. Looking at the structures of the two compounds, I would guess that both of them are demethylated in vivo (like diclofop-methyl). If so, the molecules that reach the target are the demethylated versions of these compounds. If I am right, the I50 values of the demethylated compounds on the enzyme will be much lower that what is reported for the O-methylated forms, and the I50 values of the demethylated compounds as herbicides on Arabidopsis in agar will be much higher than that of the O-methylated forms. These experiments should be done in future work, and this possibility should be discussed in the paper because it will be obvious to most scientists who deal with herbicide chemistry and mode of action.

We thank the reviewer for the insightful feedback. The chemistry of the methyl ester of diclofop-methyl and the methyl ethers of our compounds is different, but the proposed demethylation is an interesting hypothesis. We have now included this in the Discussion as per response 1 above.

Reviewer #2:[…] Weaknesses:Some questions remain about the potential utility of the identified inhibitors as effective herbicides. The soil application experiment showed biological activity by reducing plant growth, but the possible use of the inhibitors for post-emergence weed control was not explored. The inhibitors would likely be non-selective, meaning they would also damage crops, so the potential for generating resistant crops by identifying metabolic detoxification genes from bacteria could be explored.Comments for the authors:This is an outstanding manuscript. I have a few comments for some points that could be addressed to further improve the paper.Regarding the soil experiment, was the compound sprayed onto the soil before placing the seeds, or was the compound sprayed directly on the seeds? Were seeds covered by soil, whether before or after spraying, and if covered, to what depth of soil? This is important to know if the seedlings are absorbing the herbicide from the soil-water solution, or if they were contacted directly by the spray droplets at application.

We thank the reviewer for their time and feedback. The seeds were placed in an indentation in wet soil, then the compound was pipetted directly onto the seeds before covering with moist soil. The compound mixture was pipetted directly onto seeds and then moist soil was filled in. The depth of the seeds was ~1 cm from the surface. We believe that the seeds were likely to absorb the compound mixture directly through droplet contact.

We have amended the Material and Methods section as described below:

Lines 506-510: “Subsequently, ~30 seeds were sown into a small indentation (depth ~1 cm) in moist seed raising soil (pH 5.5) (Biogro, Dandenong South, VIC, Australia), supplemented with 0.22% (w/w) Nutricote N12 Micro 140 day-controlled release fertiliser (Yates, Sydney, NSW, Australia). Seeds were treated with 1 mL (equivalent to 1200 g·ha^-1^) of compound or vehicle control by pipetting and covered with soil just prior to transfer to a CER and images taken after 7 days.”

A few more details needed on the soil, such as composition (pH; percent sand, silt, and clay; percent organic matter).

Please see our response above.

Regarding the application of the herbicide solution to the soil, the authors state that 1 mL was applied, but how? Was it pipetted on or sprayed? The 1 mL volume is quite a lot for the small area, and may represent an unrealistically high quantity.

As described in our first response to reviewer #2, the 1 mL of compound mixture applied was pipetted directly onto the seeds. We agree that this is a high quantity and likely not applicable for the field. However, this manuscript provides proof-of-concept data for DHDPS inhibition and is intended to lay the foundations for inhibitors with greater potency, which will likely require a lower application to achieve phytotoxicity.

Also need to state volume of solution sprayed on soil (the solution was 300 mg/L of the herbicide but what volume of this was applied to the soil?) This is important to enable replication and testing by others.

The results in Figure 8 are from a single 1 mL dosage of the compound mixtureWe have added the dosage in g/ha as well to make the information more universally applicable.

The authors state that there are two copies of the DHDPS gene in plants; suggest that they consider reporting published Arabidopsis RNA seq data for these genes to state what is their relative expression level, whether relatively low, mid, or high? And is there any tissue or developmental specificity known for these genes? These questions are relevant because an ideal herbicide target has lower expression, thereby making it easier to inhibit.

We thank the reviewer for this helpful suggestion. We have added the following to the Introduction:

Lines 103-112: “RNA sequencing data have elucidated that both *DHDPS* encoding genes are expressed at all stages of *A. thaliana* development, with the most prominent expression at the seed development stages, in the dry seeds and during germination (Klepikova et al., 2016). […] This is a key consideration in target validation as low expressing targets will require less inhibitor to achieve phytotoxicity.”

This herbicide would apparently be non-selective in crops, given that it has conserved sequence in rice and Arabidopsis, grape, wheat. The authors could perhaps explore a few more dicot plant families containing important weeds to determine if sequence conservation is maintained.

We have added DHDPS sequences for two dicotyledonous plants, *Nicotiana tabacum* and *Solanum lycopersicum*, to sequence alignments presented in Figure 2C, Figure 2E, Figure 5D and Figure 2—figure supplement 1.

Also there is the question of how this herbicide would be used in crops; if non-selective, a resistance transgene would be needed. The authors could consider proposing in the discussion to run bacteria screens to look for metabolism genes to engineer crops?

We would like to note that bacterial screening would be difficult as the compounds presented in the manuscript have no antibacterial activity. However, we agree that if the compounds were improved to the point that they could be used in agricultural settings, resistance would need to be engineered into crops. Accordingly, we have added the following to the Discussion:

Lines 376-382: “To further pursue the compounds described herein and eventually deploy more potent analogues as herbicides, tolerant crops will eventually need to be engineered. […] Given that the inhibitors do not bind at the active site, it should be possible to engineer tolerant crops with mutant DHDPS enzymes that are not susceptible to inhibition with minimal effects on enzyme function.”

Regarding how the herbicide would be used, what about leaf uptake?

Leaf uptake would require formulation optimisation. Based on the potency of the current generation of compounds described in the manuscript, pre-emergence is likely the most efficacious route. We are mindful that the compounds presented in this manuscript are not yet potent enough to optimise delivery, as would be required for the development of a herbicide mixture.

What are the physicochemical properties of the compounds?

We have added the requested information to Supplementary File 1.

Predictions about binding to soil? Consider running them through a predictor to report log Kow, any pKa that would be relevant for uptake and acid trapping, and try to predict if these would translocate or not. A potentially helpful reference for this would be:Gandy, M.N., Corral, M.G., Mylne, J.S. and Stubbs, K.A., 2015. An interactive database to explore herbicide physicochemical properties. Organic & biomolecular chemistry, 13(20), pp.5586-5590.

Please refer to our response above, where clog*P* (KOW) and pKa values have been added to Supplementary File 1. These compounds are weak acids, which is likely to influence translocation across the cell wall and the potential of ion trapping within the alkaline compartments of the cell. Similarly, alkaline soil is likely to result in more binding than acidic soil, as we have used in this study (please see our first response to reviewer #2). Testing the ability of our compounds to bind to soil would be of interest, however, this is beyond the scope of this manuscript.

Line 345-346: hydrophilicity is not generally expected to increase movement across cell wall, should this be hydrophobicity? Again the log Kow to indicate water solubility should be reported.

We have amended the following to reflect the point made by the reviewer:

Lines 371-375: “It would also be of interest to study inhibitors with increased hydrophobicity and thus, potentially enhanced transport through the epidermis, cell wall and plastid membrane, to reach the DHDPS target. However, it is important to recognise that there is a fine balance between membrane permeability and aqueous solubility that must be achieved for any new herbicides.”

In addition to the cell wall as a barrier, herbicides must pass the cell membrane and if plastid targeted, the plastid membrane, along with passing the wax and cuticle on the epidermis; consider revising this discussion point to expand that the cell wall is not the only barrier for uptake.

We have amended the line as described in our response above.

Reviewer #3:[…] This study investigates dihydrodipicolinate synthase (the rate limiting step in lysine biosynthesis as a potential herbicide target site) and identifies a couple of molecules that provide some level of inhibition. Generally speaking this study is done properly and the experiments appear to have been done correctly. I have made several grammatical/typographical suggestions directly on the manuscript.First, the authors do a good job demonstrating that DHDPS can be inhibited, establishing the fact that it might be a suitable target site. The two most active molecules are not very potent, so I am glad the authors don't oversell the potency of the molecules they discovered. However, it is interesting that these compounds have some soil activity, albeit at very high concentrations.I find the portion of the manuscript characterizing the binding of these inhibitors could be developed a little more.1. At first glance, I am not sure if the pose of the inhibitors shown in Figure 5 is a true reflection of their binding on DHDPS. These poses may be a crystallization artifact. The inhibitors appear to interact with each other via pi pi stacking and I wonder if they would crystalize like this if by themselves.2. The authors did note that some histidine residues moved to accommodate for the presence of the inhibitors. However, one would expect that the inhibitors binding would be stabilized by pi pi stacking with the histidine or the tryptophan residues within the proximity of the molecules, but this does not appear to be the case. May be the authors should discuss this.

We thank the reviewer for their time and feedback. We do not believe that the stacking is a crystal artefact, rather that it is a true mode of binding. This is evidenced by the inhibitor molecules located on the surface, where they form pi stacks with residues, rather than each other. Moreover, we expected these compounds to be allosteric inhibitors of DHDPS and they bind in the expected region, albeit not exactly where lysine binds. The most notable residue movements were of D117 and W116. Interestingly, the inhibitor molecules force D117 to adopt a different conformation, which in turn causes W116 to adopt a new conformation. This is described in lines 213-215. We agree that these inhibitors would likely form a pi pi stack with the aromatic residues in the near vicinity. However, the residues W116 and H150 are not positioned to allow for this, instead, these residues interact with the methoxy group. To illustrate non-specific binding between symmetry mates, we have included a supplementary figure (Figure 5—figure supplement 1). The figure shows that inhibitor interactions at the surface of the protein are exclusive to where protein molecules interact with one another, specifically, chains B and D. For chains A and C, there are no interacting symmetry mates and we see no inhibitors wedged between protein molecules. Based on this and the proximity of the four inhibitor molecules at the centre of the protein to the lysine binding site, we are confident that we are demonstrating the true site of inhibition.

3. The binding of these inhibitors appears to overlap with the allosteric binding site of lysine. This may explain why the potency of the inhibitors is not very high. To my knowledge, no commercial herbicide is known to target an allosteric site. Their may be come biochemical reasons for this. Allosteric regulation by inhibitors may not be 'lethal' enough. The authors should discuss this possibility. I suggest that they also include a statement indicating that other inhibitors interacting with the catalytic domain of DHDPS might lead to more active herbicide leads.

We thank the reviewer for the feedback. We have added the following to the Discussion to address this point:

Lines 325-334: “Several iterations of active site inhibitors against bacterial DHDPS enzymes have been developed, but most of these have, at best, high micromolar potency (Christoff et al., 2019). […] Although herbicides commonly directly target, or bind closely to, the active site of their respective enzyme target, this study shows allosteric site inhibitors can be employed and afford lethal phytotoxicity.”

The authors will not have any problem addressing the grammatical/typographical suggestions I made directly on the manuscript (see attached PDF).

We thank the reviewer for their notes and have made the appropriate changes.